# Deep Convolutional Neural Network for Image Deconvolution

**Li Xu** *
Lenovo Research & Technology
xulihk@lenovo.com

**Jimmy SJ. Ren**
Lenovo Research & Technology
jimmy.sj.ren@gmail.com

**Ce Liu**
Microsoft Research
celiu@microsoft.com

**Jiaya Jia**
The Chinese University of Hong Kong
leojia@cse.cuhk.edu.hk

## Abstract

Many fundamental image-related problems involve deconvolution operators. Real blur degradation seldom complies with an ideal linear convolution model due to camera noise, saturation, image compression, to name a few. Instead of perfectly modeling outliers, which is rather challenging from a generative model perspective, we develop a deep convolutional neural network to capture the characteristics of degradation. We note directly applying existing deep neural networks does not produce reasonable results. Our solution is to establish the connection between traditional optimization-based schemes and a neural network architecture where a novel, separable structure is introduced as a reliable support for robust deconvolution against artifacts. Our network contains two submodules, both trained in a supervised manner with proper initialization. They yield decent performance on non-blind image deconvolution compared to previous generative-model based methods.

## 1   Introduction

Many image and video degradation processes can be modeled as translation-invariant convolution. To restore these visual data, the inverse process, i.e., *deconvolution*, becomes a vital tool in motion deblurring [1, 2, 3, 4], super-resolution [5, 6], and extended depth of field [7].

In applications involving images captured by cameras, outliers such as saturation, limited image boundary, noise, or compression artifacts are unavoidable. Previous research has shown that improperly handling these problems could raise a broad set of artifacts related to image content, which are very difficult to remove. So there was work dedicated to modeling and addressing each particular type of artifacts in non-blind deconvolution for suppressing ringing artifacts [8], removing noise [9], and dealing with saturated regions [9, 10]. These methods can be further refined by incorporating patch-level statistics [11] or other schemes [4]. Because each method has its own specialty as well as limitation, there is no solution yet to uniformly address all these issues. One example is shown in Fig. 1 – a partially saturated blur image with compression errors can already fail many existing approaches.

One possibility to remove these artifacts is via employing generative models. However, these models are usually made upon strong assumptions, such as identical and independently distributed noise, which may not hold for real images. This accounts for the fact that even advanced algorithms can be affected when the image blur properties are slightly changed.

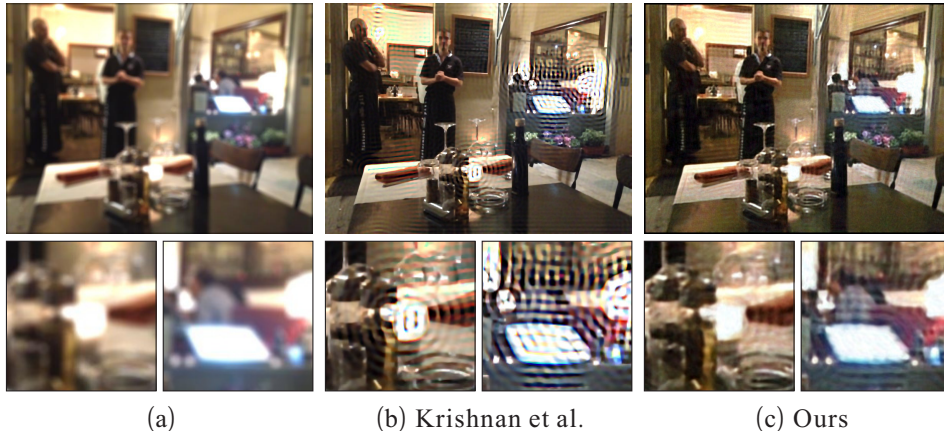

| (a) | (b) Krishnan et al. | (c) Ours |

Figure 1: A challenging deconvolution example. (a) is the blurry input with partially saturated regions. (b) is the result of [3] using hyper-Laplacian prior. (c) is our result.

In this paper, we initiate the procedure for natural image deconvolution not based on their physically or mathematically based characteristics. Instead, we show a new direction to build a data-driven system using image samples that can be easily produced from cameras or collected online.

We use the convolutional neural network (CNN) to learn the deconvolution operation without the need to know the cause of visual artifacts. We also do not rely on any pre-process to deblur the image, unlike previous learning based approaches [12, 13]. In fact, it is non-trivial to find a proper network architecture for deconvolution. Previous de-noise neural network [14, 15, 16] cannot be directly adopted since deconvolution may involve many neighboring pixels and result in a very complex energy function with nonlinear degradation. This makes parameter learning quite challenging.

In our work, we bridge the gap between an empirically-determined convolutional neural network and existing approaches with generative models in the context of pseudo-inverse of deconvolution. It enables a practical system and, more importantly, provides an empirically effective strategy to initialize the weights in the network, which otherwise cannot be easily obtained in the conventional random-initialization training procedure. Experiments show that our system outperforms previous ones especially when the blurred input images are partially saturated.

## 2   Related Work

Deconvolution was studied in different fields due to its fundamentality in image restoration. Most previous methods tackle the problem from a generative perspective assuming known image noise model and natural image gradients following certain distributions.

In the Richardson-Lucy method [17], image noise is assumed to follow a Poisson distribution. Wiener Deconvolution [18] imposes equivalent Gaussian assumption for both noise and image gradients. These early approaches suffer from overly smoothed edges and ringing artifacts.

Recent development on deconvolution shows that regularization terms with sparse image priors are important to preserve sharp edges and suppress artifacts. The sparse image priors follow heavy-tailed distributions, such as a Gaussian Mixture Model [1, 11] or a hyper-Laplacian [7, 3], which could be efficiently optimized using half-quadratic (HQ) splitting [3]. To capture image statistics with larger spatial support, the energy is further modeled within a Conditional Random Field (CRF) framework [19] and on image patches [11]. While the last step of HQ method is quadratic optimization, Schmidt et al. [4] showed that it is possible to directly train a Gaussian CRF from synthetic blur data.

To handle outliers such as saturation, Cho et al. [9] used variational EM to exclude outlier regions from a Gaussian likelihood. Whyte et al. [10] introduced an auxiliary variable in the Richardson-Lucy method. An explicit denoise pass is added to deconvolution, where the denoise approach is carefully engineered [20] or trained from noisy data [12]. The generative approaches typically have difficulties to handle complex outliers that are not independent and identically distributed.

Another trend for image restoration is to leverage the deep neural network structure and big data to train the restoration function. The degradation is therefore no longer limited to one model regarding image noise. Burger et al. [14] showed that the plain multi-layer perceptrons can produce decent results and handle different types of noise. Xie et al. [15] showed that a stacked denoise autoencoder (SDAE) structure [21] is a good choice for denoise and inpainting. Agostinelli et al. [22] generalized it by combining multiple SDAE for handling different types of noise. In [23] and [16], the convolutional neural network (CNN) architecture [24] was used to handle strong noise such as raindrop and lens dirt. Schuler et al. [13] added MLPs to a direct deconvolution to remove artifacts. Though the network structure works well for denoise, it does not work similarly for deconvolution. How to adapt the architecture is the main problem to address in this paper.

## 3 Blur Degradation

We consider real-world image blur that suffers from several types of degradation including clipped intensity (saturation), camera noise, and compression artifacts. The blur model is given by

$$\hat{y} = \psi_b[\phi(\alpha x * k + n)], \tag{1}$$

where $\alpha x$ represents the latent sharp image. The notation $\alpha \geq 1$ is to indicate the fact that $\alpha x$ could have values exceeding the dynamic range of camera sensors and thus be clipped. $k$ is the known convolution kernel, or typically referred to as a point spread function (PSF), $n$ models additive camera noise. $\phi(\cdot)$ is a clipping function to model saturation, defined as $\phi(z) = \min(z, z_{max})$, where $z_{max}$ is a range threshold. $\psi_b[\cdot]$ is a nonlinear (e.g., JPEG) compression operator.

We note that even with $\hat{y}$ and kernel $k$, restoring $\alpha x$ is intractable, simply because the information loss caused by clipping. In this regard, our goal is to restore the clipped input $\hat{x}$, where $\hat{x} = \phi(\alpha x)$.

Although solving for $\hat{x}$ with a complex energy function that involves Eq. (1) is difficult, the generation of blurry image from an input $x$ is quite straightforward by image synthesis according to the convolution model taking all kinds of possible image degradation into generation. This motivates a learning procedure for deconvolution, using training image pairs $\{\hat{x}_i, \hat{y}_i\}$, where index $i \in N$.

## 4 Analysis

The goal is to train a network architecture $f(\cdot)$ that minimizes

$$\frac{1}{2|N|} \sum_{i \in N} \|f(\hat{y}_i) - \hat{x}_i\|^2, \tag{2}$$

where $|N|$ is the number of image pairs in the sample set.

We have used the recent two deep neural networks to solve this problem, but failed. One is the Stacked Sparse Denoise Autoencoder (SSDAE) [15] and the other is the convolutional neural network (CNN) used in [16]. Both of them are designed for image denoise. For SSDAE, we use patch size $17 \times 17$ as suggested in [14]. The CNN implementation is provided by the authors of [16]. We collect two million sharp patches together with their blurred versions in training.

One example is shown in Fig. 2 where (a) is a blurred image. Fig. 2(b) and (c) show the results of SSDAE and CNN. The result of SSDAE in (b) is still blurry. The CNN structure works relatively better. But it suffers from remaining blurry edges and strong ghosting artifacts. This is because these network structures are for denoise and do not consider necessary deconvolution properties. More explanations are provided from a generative perspective in what follows.

### 4.1 Pseudo Inverse Kernels

The deconvolution task can be approximated by a convolutional network by nature. We consider the following simple linear blur model

$$y = x * k.$$

The spatial convolution can be transformed to a frequency domain multiplication, yielding

$$\mathcal{F}(y) = \mathcal{F}(x) \cdot \mathcal{F}(k).$$

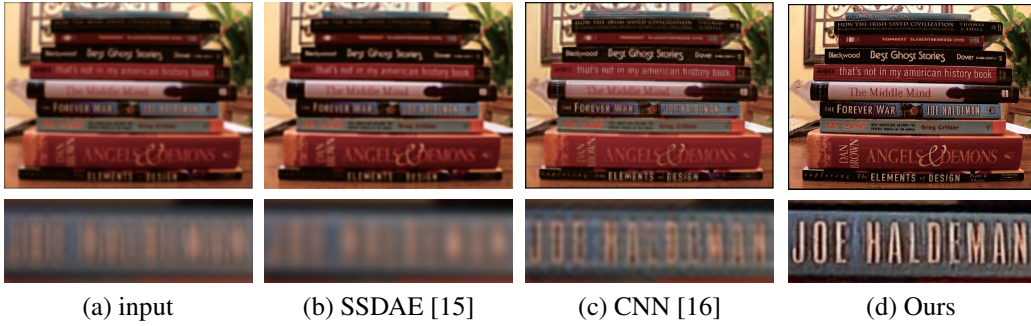

|  (a) input | (b) SSDAE [15] | (c) CNN [16] | (d) Ours |

Figure 2: Existing stacked denoise autoencoder and convolutional neural network structures cannot solve the deconvolution problem.

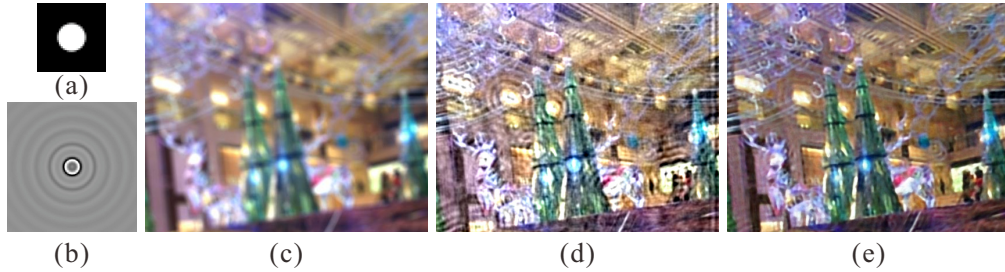

Figure 3: Pseudo inverse kernel and deconvolution examples.

$\mathcal{F}(\cdot)$ denotes the discrete Fourier transform (DFT). Operator $\cdot$ is element-wise multiplication. In Fourier domain, $x$ can be obtained as

$$x = \mathcal{F}^{-1}(\mathcal{F}(y)/\mathcal{F}(k)) = \mathcal{F}^{-1}(1/\mathcal{F}(k)) * y,$$

where $\mathcal{F}^{-1}$ is the inverse discrete Fourier transform. While the solver for $x$ is written in a form of spatial convolution with a kernel $\mathcal{F}^{-1}(1/\mathcal{F}(k))$, the kernel is actually a repetitive signal spanning the whole spatial domain without a compact support. When noise arises, regularization terms are commonly involved to avoid division-by-zero in frequency domain, which makes the pseudo inverse falls off quickly in spatial domain [25].

The classical Wiener deconvolution is equivalent to using Tikhonov regularizer [2]. The Wiener deconvolution can be expressed as

$$x = \mathcal{F}^{-1}(\frac{1}{\mathcal{F}(k)}\{\frac{|\mathcal{F}(k)|^2}{|\mathcal{F}(k)|^2 + \frac{1}{SNR}}\}) * y = k^\dagger * y,$$

where $SNR$ is the signal-to-noise ratio. $k^\dagger$ denotes the pseudo inverse kernel. Strong noise leads to a large $\frac{1}{SNR}$, which corresponds to strongly regularized inversion. We note that with the introduction of $SNR$, $k^\dagger$ becomes compact with a finite support. Fig. 3(a) shows a disk blur kernel of radius 7, which is commonly used to model focal blur. The pseudo-inverse kernel $k^\dagger$ with $SNR = 1E - 4$ is given in Fig. 3(b). A blurred image with this kernel is shown in Fig. 3(c). Deconvolution results with $k^\dagger$ are in (d). A level of blur is removed from the image. But noise and saturation cause visual artifacts, in compliance with our understanding of Wiener deconvolution.

Although the Wiener method is not state-of-the-art, its byproduct that the inverse kernel is with a finite yet large spatial support becomes vastly useful in our neural network system, which manifests that deconvolution can be well approximated by spatial convolution with sufficiently large kernels. This explains unsuccessful application of SSDA and CNN directly to deconvolution in Fig. 2 as follows.

- SSDA does not capture well the nature of convolution with its fully connected structures.
- CNN performs better since deconvolution can be approximated by large-kernel convolution as explained above.

- Previous CNN uses small convolution kernels. It is however not an appropriate configuration in our deconvolution problem.

It thus can be summarized that using deep neural networks to perform deconvolution is by no means straightforward. Simply modifying the network by employing large convolution kernels would lead to higher difficulties in training. We present a new structure to update the network in what follows. Our result in Fig. 3 is shown in (e).

## 5   Network Architecture

We transform the simple pseudo inverse kernel for deconvolution into a convolutional *network*, based on the kernel separability theorem. It makes the network more expressive with the mapping to higher dimensions to accommodate nonlinearity. This system is benefited from large training data.

### 5.1   Kernel Separability

Kernel separability is achieved via singular value decomposition (SVD) [26]. Given the inverse kernel $k^\dagger$, decomposition $k^\dagger = USV^T$ exists. We denote by $u_j$ and $v_j$ the $j^{th}$ columns of $U$ and $V$, $s_j$ the $j^{th}$ singular value. The original pseudo deconvolution can be expressed as

$$k^\dagger * y = \sum_j s_j \cdot u_j * (v_j^T * y),\qquad(3)$$

which shows 2D convolution can be deemed as a weighted sum of separable 1D filters. In practice, we can well approximate $k^\dagger$ by a small number of separable filters by dropping out kernels associated with zero or very small $s_j$. We have experimented with real blur kernels to ignore singular values smaller than 0.01. The resulting average number of separable kernels is about 30 [25]. Using a smaller $SNR$ ratio, the inverse kernel has a smaller spatial support. We also found that an inverse kernel with length 100 is typically enough to generate visually plausible deconvolution results. This is important information in designing the network architecture.

### 5.2   Image Deconvolution CNN (DCNN)

We describe our image deconvolution convolutional neural network (DCNN) based on the separable kernels. This network is expressed as

$$h_3 = W_3 * h_2;\quad h_l = \sigma(W_l * h_{l-1} + b_{l-1}),\ l \in \{1, 2\};\quad h_0 = \hat{y},$$

where $W_l$ is the weight mapping the $(l-1)^{th}$ layer to the $l^{th}$ one and $b_{l-1}$ is the vector value bias. $\sigma(\cdot)$ is the nonlinear function, which can be sigmoid or hyperbolic tangent.

Our network contains two hidden layers similar to the separable kernel inversion setting. The first hidden layer $h_1$ is generated by applying 38 large-scale one-dimensional kernels of size $121 \times 1$, according to the analysis in Section 5.1. The values 38 and 121 are empirically determined, which can be altered for different inputs. The second hidden layer $h_2$ is generated by applying $38$ $1 \times 121$ convolution kernels to each of the 38 maps in $h_1$. To generate results, a $1 \times 1 \times 38$ kernel is applied, analogous to the linear combination using singular value $s_j$.

The architecture has several advantages for deconvolution. First, it assembles separable kernel inversion for deconvolution and therefore is guaranteed to be optimal. Second, the nonlinear terms and high dimensional structure make the network more expressive than traditional pseudo-inverse. It is reasonably robust to outliers.

### 5.3   Training DCNN

The network can be trained either by random-weight initialization or by the initialization from the separable kernel inversion, since they share the exact same structure.

We experiment with both strategies on natural images, which are all degraded by additive Gaussian noise (AWG) and JPEG compression. These images are in two categories – one with strong color saturation and one without. Note saturation affects many existing deconvolution algorithms a lot.

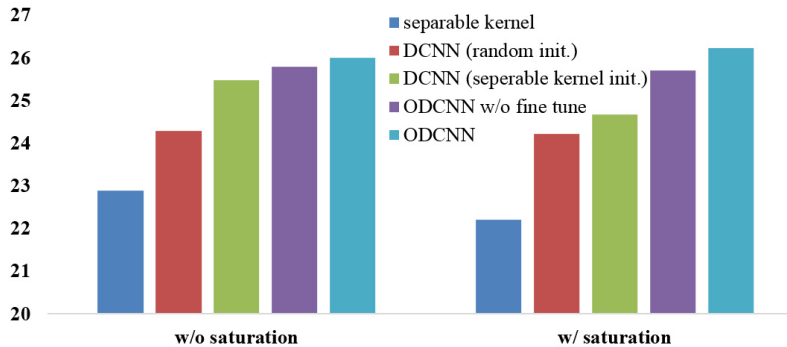

Figure 4: PSNRs produced in different stages of our convolutional neural network architecture.

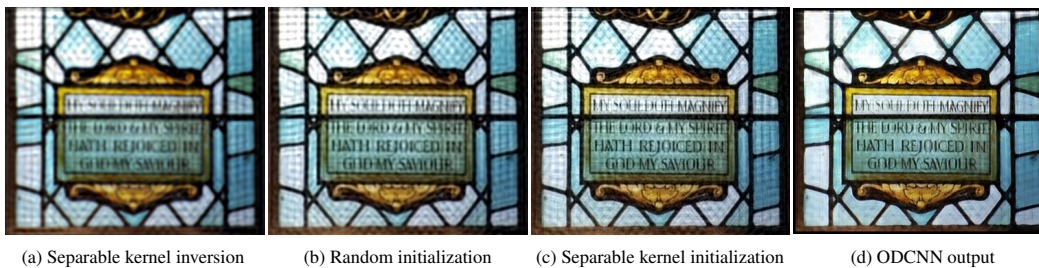

(a) Separable kernel inversion      (b) Random initialization      (c) Separable kernel initialization      (d) ODCNN output

Figure 5: Results comparisons in different stages of our deconvolution CNN.

The PSNRs are shown as the first three bars in Fig. 4. We obtain the following observations.

- The trained network has an advantage over simply performing separable kernel inversion, no matter with random initialization or initialization from pseudo-inverse. Our interpretation is that the network, with high dimensional mapping and nonlinearity, is more expressive than simple separable kernel inversion.
- The method with separable kernel inversion initialization yields higher PSNRs than that with random initialization, suggesting that initial values affect this network and thus can be tuned.

Visual comparison is provided in Fig. 5(a)-(c), where the results of separable kernel inversion, training with random weights, and of training with separable kernel inversion initialization are shown. The result in (c) obviously contains sharp edges and more details. Note that the final trained DCNN is not equivalent to any existing inverse-kernel function even with various regularization, due to the involved high-dimensional mapping with nonlinearities.

The performance of deconvolution CNN decreases for images with color saturation. Visual artifacts could also be yielded due to noise and compression. We in the next section turn to a deeper structure to address these remaining problems, by incorporating a denoise CNN module.

### 5.4 Outlier-rejection Deconvolution CNN (ODCNN)

Our complete network is formed as the concatenation of the deconvolution CNN module with a denoise CNN [16]. The overall structure is shown in Fig. 6. The denoise CNN module has two hidden layers with $512$ feature maps. The input image is convolved with $512$ kernels of size $16 \times 16$ to be fed into the hidden layer.

The two network modules are concatenated in our system by combining the last layer of deconvolution CNN with the input of denoise CNN. This is done by merging the $1 \times 1 \times 36$ kernel with 512 $16 \times 16$ kernels to generate 512 kernels of size $16 \times 16 \times 36$. Note that there is no nonlinearity when combining the two modules. While the number of weights grows due to the merge, it allows for a flexible procedure and achieves decent performance, by further incorporating fine tuning.

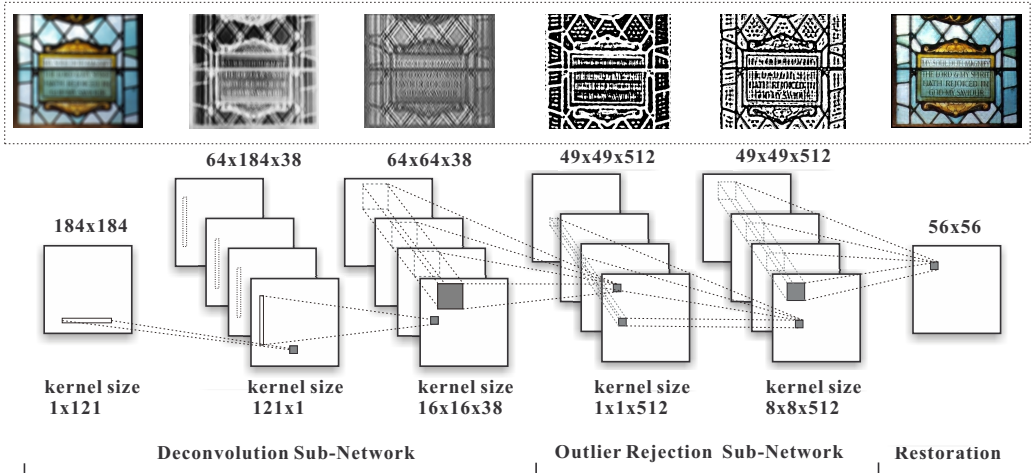

Figure 6: Our complete network architecture for deep deconvolution.

## 5.5 Training ODCNN

We blur natural images for training – thus it is easy to obtain a large number of data. Specifically, we use 2,500 natural images downloaded from Flickr. Two million patches are randomly sampled from them. Concatenating the two network modules can describe the deconvolution process and enhance the ability to suppress unwanted structures. We train the sub-networks separately. The deconvolution CNN is trained using the initialization from separable inversion as described before. The output of deconvolution CNN is then taken as the input of the denoise CNN.

Fine tuning is performed by feeding one hundred thousand $184 \times 184$ patches into the whole network. The training samples contain all patches possibly with noise, saturation, and compression artifacts. The statistics of adding denoise CNN are also plotted in Fig. 4. The outlier-rejection CNN after fine tuning improves the overall performance up to 2dB, especially for those saturated regions.

## 6 More Discussions

Our approach differs from previous ones in several ways. First, we identify the necessity of using a relatively large kernel support for convolutional neural network to deal with deconvolution. To avoid rapid weight-size expansion, we advocate the use of 1D kernels. Second, we propose a supervised pre-training on the sub-network that corresponds to reinterpretation of Wiener deconvolution. Third, we apply traditional deconvolution to network initialization, where generative solvers can guide neural network learning and significantly improve performance.

Fig. 6 shows that a new convolutional neural network architecture is capable of dealing with deconvolution. Without a good understanding of the functionality of each sub-net and performing supervised pre-training, however, it is difficult to make the network work very well. Training the whole network with random initialization is less preferred because the training algorithm stops halfway without further energy reduction. The corresponding results are similarly blurry as the input images. To understand it, we visualize intermediate results from the deconvolutional CNN sub-network, which generates 38 intermediate maps. The results are shown in Fig. 7, where (a) is the selected three results obtained by random-initialization training and (b) is the results at the corresponding nodes from our better-initialized process. The maps in (a) look like the high-frequency part of the blurry input, indicating random initialization is likely to generate high-pass filters. Without proper starting values, its chance is very small to reach the component maps shown in (b) where sharper edges present, fully usable for further denoise and artifact removal.

Zeiler et al. [27] showed that sparsely regularized deconvolution can be used to extract useful middle-level representation in their deconvolution network. Our deconvolution CNN can be used to approximate this structure, unifying the process in a deeper convolutional neural network.

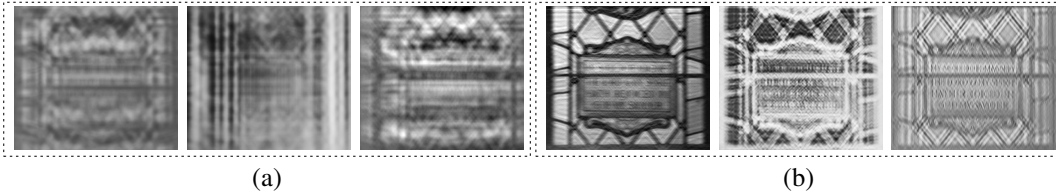

|        (a)          |         (b)          |

Figure 7: Comparisons of intermediate results from deconvolution CNN. (a) Maps from random initialization. (b) More informative maps with our initialization scheme.

| kernel type | Krishnan [3] | Levin [7] | Cho [9] | Whyte [10] | Schuler [13] | Schmidt [4] | Ours |
|---|---|---|---|---|---|---|---|
| disk sat. | 24.05dB | 24.44dB | 25.35dB | 24.47dB | 23.14dB | 24.01dB | **26.23dB** |
| disk | 25.94dB | 24.54dB | 23.97dB | 22.84dB | 24.67dB | 24.71dB | **26.01dB** |
| motion sat. | 24.07dB | 23.58dB | 25.65 dB | 25.54dB | 24.92dB | 25.33dB | **27.76dB** |
| motion | 25.07dB | 24.47 dB | 24.29dB | 23.65dB | 25.27dB | 25.49dB | **27.92dB** |

Table 1: Quantitative comparison on the evaluation image set.

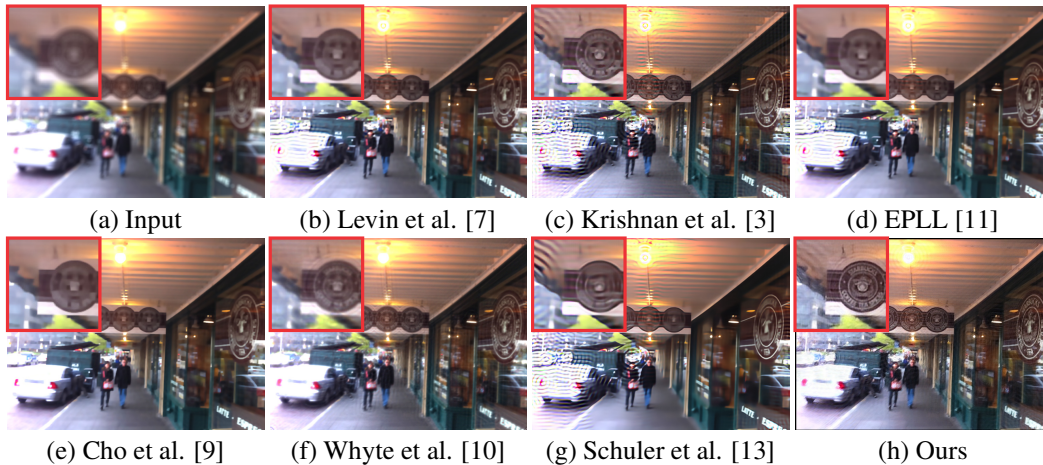

| (a) Input | (b) Levin et al. [7] | (c) Krishnan et al. [3] | (d) EPLL [11] |
|---|---|---|---|
| (e) Cho et al. [9] | (f) Whyte et al. [10] | (g) Schuler et al. [13] | (h) Ours |

Figure 8: Visual comparison of deconvolution results.

## 7  Experiments and Conclusion

We have presented several deconvolution results. Here we show quantitative evaluation of our method against state-of-the-art approaches, including sparse prior deconvolution [7], hyper-Laplacian prior method [3], variational EM for outliers [9], saturation-aware approach [10], learning based approach [13] and the discriminative approach [4]. We compare performance using both disk and motion kernels. The average PSNRs are listed in Table 1. Fig. 8 shows a visual comparison. Our method achieves decent results quantitatively and visually. The implementation, as well as the dataset, is available at the project webpage.

To conclude this paper, we have proposed a new deep convolutional network structure for the challenging image deconvolution task. Our main contribution is to let traditional deconvolution schemes guide neural networks and approximate deconvolution by a series of convolution steps. Our system novelly uses two modules corresponding to deconvolution and artifact removal. While the network is difficult to train as a whole, we adopt two supervised pre-training steps to initialize sub-networks. High-quality deconvolution results bear out the effectiveness of this approach.

## Footnotes

*Project webpage: http://www.lxu.me/projects/dcnn/. The paper is partially supported by a grant from the Research Grants Council of the Hong Kong Special Administrative Region (Project No. 413113).

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
