[Reviews · NeurIPS 2014]

Submitted by Assigned_Reviewer_19

The paper proposes to use a deep convolutional neural network for denoising images by generating a lot of noisy and noiseless image pairs using a synthetic blurring process. The proposed method achieves good results on a number of image deblurring tasks.

The idea is simple and elegant. It is observed that a 2D deconvolution, which is an inverse of the convolution operator, is itself a 2D convolution operator, albeit with a very large support. The proposed conv net uses large 2D separable convolutions, which can be efficiently implemented and learned. This idea is simple, elegant, and appears to work. The qualitative results are impressive.

I didn’t understand the precise details of the pre-training, as I was under the impression that the entire system is trained jointly. I also couldn’t tell whether table 1 was obtained by re-implementing the methods of others’, or by citing their published results. For example, I couldn’t find the results in Krishnan’s [3] paper. Also, I am concerned about the extremely small size of the test set (16 images), which suggests that the results may be very noisy.
Summary: A new simple convolutional method for denoising is proposed. It has a new idea: deconvolution = convolution with a large support, so large 2D separable filters in a big conv net will help it denoise better. The method appears to work. I am unclear about some aspects of training, and concerned about the small size of the test set.

Submitted by Assigned_Reviewer_25

This paper applies deep CNN for the problem of image deconvolution. It extends the success of deep neural network from image denoising to image devconvolution by using separable kernels and outlier rejection. Its results outperform previous works in [3,7,9,14].

However, this paper seems to overlook two recent papers:
1 Schuler et al, A machine learning approach for non-blind image deconvolution, CVPR13
2 Zeiler et al, Deconvolutional Networks, CVPR10
It will be good to compare with the performance in these papers.
Summary: This paper discusses how to design deep CNN for the problem of image deconvolution and outperforms a number of previous works.

Submitted by Assigned_Reviewer_44

This paper presents a method for nonblind deconvolution of blurry images, that also can also fix artifacts (e.g. compression, clipping) in the input, and is robust to deviations from the input generation model. A convolutional network is used both to deblur and fix artifacts; deblurring is performed using a sequence of horizontal and vertical conv kernels, taking advantage of a high degree of separability in the pseudoinverse blur kernel, and are initialized with a decomposition of the pseudoinverse. A standard compact-kernel convnet is stacked on top, allowing further fixing of artifacts and noise, and traned end-to-end with pairs of blurry and ground truth images.

This is a nice approach that appears to obtain good results on the examples presented. However, I feel the data could be better explained, and perhaps the test set expanded in size. I didn't see any description of the sources for the training or testing data -- where did these images come from? How were the blurry versions generated (in both train or test), and what kernels, noise, etc. were used exactly? I feel this is a significant gap, and hope it might be addressed in the authors' rebuttal.

Questions/comments:

* l.150 "by taking a great number of pictures" -- I'm not sure I understand this, could you explain a bit more?

* l.166 "two million sharp patches" -- from where were these drawn?

* l.198: description of figure does not match the figure itself (there is just one example instead of two and the letters do not match)

* l.214: extending with large conv kernels: An alternative might be an FFT conv -- maybe this could be touched on in the text a bit?

* l.262: Is the nonlinearity sigma applied between horizontal and vertical conv layers? It seems that it is. If so, the operations are starting to differ from a linear decomposition of the original pseudoinverse convolution. Is this good or bad in this case? What if this nonlinearity is removed?

* If there is a nonlinearity between horizontal and vertical, is there any benefit to stacking another layer of horizontal and vertical convs (with nonlin), and tweak the initialization either to stretch the pseudoinverse over all these applications or make the second set of kernels start from random?

* l.268 "high-dimensional structure make the network more expressive than traditional pseudo-inverse and reasonably robust to outliers": Which high dimensions are you referring to here? Is it the 36 maps from the head of the SVD? If so, would response of the largest eigenvectors potentially be truncated by the sigmoid? Or does the network learn to work around this by moving away from the decomposition, and this is what makes it robust?

* Likewise, I'm curious what might happen by initializing by duplicating the largest eigenvectors multiple times and adding a bit of gaussian noise to the weights?

* l.306 "16 images" -- this seems a bit small for a test set, though I'm not sure how large the images are; perhaps this could be augmented and/or explained a bit more?

* l.342 "done by merging" kernels: The merged kernel is not quite the same network as performing each in sequence: there are more params. But this probably is a benefit, allowing the network to perform even better, I suspect.

* l.364 "This is the first attempt to make use of generative solvers to guide neural network learning" -- I don't think this is true, and think this sentence probably should be removed. E.g. naive bayes corresponds to logistic regression; sparse coding informed the design of learned ISTA "Learning Fast Approximations of Sparse Coding" Gregor 2010; unrolled mean-field inference of RBMs in "Multi-prediction deep Boltzmann machines" Godfellow 2013.

* sec 7: Was the network retrained for each blur kernel in the experiments?
Summary: This is a nice approach with seemingly good results on the examples presented, but I feel suffers from a lack of detail regarding datasources and experiments.
Author Feedback
Author rebuttal: We thank all the reviewers for their very constructive reviews.

---
Reviewer 19

Q. Precise details of supervised pre-training
A. Our supervised pre-training contains two stages for network construction. First, we initialize weights of the deconvolution CNN as described in the paper, and train this network with blurry/clear image pairs in a supervised fashion. Second, after training this network, we treat it as a generator to produce deconvolved (but with outliers) images, which, together with their ground truth clear versions, are used to train the outlier-rejection deconvolution CNN. These two networks are combined by removing the output map of the deconvolution CNN and expanding the weights between the two networks. Finally, we fine tune the whole architecture (figure 5 in the paper).

Q. Statistics in table 1
A. The statistics on the dataset are obtained using the authors’ implementation or publicly accessible software.

Q. Size of the test set
A. The performance of different methods on the 16 images is consistent. We feel that the size is reasonable to get some meaningful statistics, while we also plan to have a larger test set in the future. We will release our implementation and image dataset to encourage comparisons.

---
Reviewer 25

Q. Two recent papers
A. We will cite and compare with them.

---
Reviewer 44

Q. Explain the data, where does the training/testing data come from? "Two million sharp patches" -- from where were these drawn?
A. We choose to blur natural images as training samples, for its simplicity to obtain a large set of training pairs. Specifically, we use 2,500 natural images downloaded from Flickr, including photos taken at day time and night time, of indoor and outdoor scenes, and with and without flash. The two million patches are randomly sampled from them.

Q. How were the blurry versions generated (in both train and test), what kernels, what noises were used
A. Data are generated according to Eq. (1) - training patches were generated by convolving clear patches with blur kernels and adding degradations. Tested kernels include motion kernels [2] and focal blur kernels (disk kernel). Additive Gaussian Noise (AWG) is added to the images.

Q. l.150 "by taking a great number of pictures" -- explain a bit more?
A. We meant a way to obtain clear/blurry image pairs is to take pictures of planner objects with noisy dotted patterns around them for recording kernels – the same method as described in [2].

Q. l.262: Is the nonlinearity sigma applied between horizontal and vertical conv layers? Is this good or bad in this case? What if this nonlinearity is removed?
A. Yes, the nonlinearity is applied between horizontal and vertical convolution layers. Only the nonlinearity between two sub-nets is omitted for combination. We did not test the network without nonlinearity yet. But based on current results, the deconvolution CNN performs better than the original pseudo inversion in terms of noise resistance, possibly owing to the nonlinearity terms.

Q. l.268 Are high dimensions referred to the 36 maps from the head of the SVD? If so, would response of the largest eigenvectors potentially be truncated by the sigmoid? Or does the network learn to work around this by moving away from the decomposition, and this is what makes it robust?
A. Yes, the 36 feature maps associated with large eigenvalues are referred to. Pre-training of decovolution CNN does make the network deviate from the original decomposition results and perform better for noisy input. So we think the sigmoid function plays an important role here.

Q. l.306 "16 images" -- this seems a bit small for a test set
A. The performance of different methods on the 16 images is consistent. We feel that the size is reasonable to get some meaningful statistics, while we also plan to have a larger test set in the future.

Q. l.342 "done by merging" kernels: The merged kernel is not quite the same network as performing each in sequence: there are more params. But this probably is a benefit, allowing the network to perform even better, I suspect.
A. We agree that this is a benefit.

Q. l.364 This sentence probably should be removed.
A. Will do.

Q. sec 7: Was the network retrained for each blur kernel in the experiments?
A. Yes, for each blur kernel we train a separate model. In practice, the outlier-rejection CNN does not need to be pre-trained again for different kernels.